# Simultaneous Prediction of Soil Properties Using Multi_CNN Model

**DOI:** 10.3390/s20216271

**Published:** 2020-11-03

**Authors:** Ruixue Li, Bo Yin, Yanping Cong, Zehua Du

**Affiliations:** 1College of Information Science and Engineering, Ocean University of China, Qingdao 266000, China; liruixue@stu.ouc.edu.cn (R.L.); congyp@ouc.edu.cn (Y.C.); duzehua@stu.ouc.edu.cn (Z.D.); 2Pilot National Laboratory for Marine Science and Technology, Qingdao 266000, China

**Keywords:** soil, vis-NIR spectroscopy, deep learning, convolutional neural networks, multi-task learning

## Abstract

Soil nutrient prediction based on near-infrared spectroscopy has become the main research direction for rapid acquisition of soil information. The development of deep learning has greatly improved the prediction accuracy of traditional modeling methods. In view of the low efficiency and low accuracy of current soil prediction models, this paper proposes a soil multi-attribute intelligent prediction method based on convolutional neural networks, by constructing a dual-stream convolutional neural network model Multi_CNN that combines one-dimensional convolution and two-dimensional convolution, the intelligent prediction of soil multi-attribute is realized. The model extracts the characteristics of soil attributes from spectral sequences and spectrograms respectively, and multiple attributes can be predicted simultaneously by feature fusion. The model is based on two different-scale soil near-infrared spectroscopy data sets for multi-attribute prediction. The experimental results show that the $R_{P}^{2}$ of the three attributes of Total Carbon, Total Nitrogen, and Alkaline Nitrogen on the small dataset are 0.94, 0.95, 0.87, respectively, and the $R_{P}^{2}$ of the attributes of Organic Carbon, Nitrogen, and Clay on the LUCAS dataset are, respectively, 0.95, 0.91, 0.83, And compared with traditional regression models and new prediction methods commonly used in soil nutrient prediction, the multi-task model proposed in this paper is more accurate.

## 1. Introduction

Soil is an important natural resource. The rapid acquisition of soil property content and spatial distribution is of great value and significance to agriculture and global change. However, the collection of soil samples consumes a large amount of cost, so the prediction of soil nutrient content has become a hot topic in soil research. Visible light near infrared (Vis-NIR) spectroscopy analysis, with its unique advantages such as rapid detection, non-destructive, non-polluting, and real-time detection, has extensive research and application foundations in soil nutrient content prediction [1,2,3]. However, the spectral data is susceptible to interference from stray light, noise, baseline drift and other factors, which affect the modeling effect. Therefore, it is necessary to preprocess the spectral data before modeling to improve the predictive ability and robustness of the model. Due to the complex characteristics of spectral data, although traditional mathematical modeling methods can perform a certain degree of analysis and prediction, its more accurate and more universal prediction process faces technical bottlenecks. With the development of machine learning, many new spectral model regression prediction algorithms have been continuously proposed and applied [4,5,6]. However, compared with traditional mathematical modeling and machine learning methods, neural network models have higher computational efficiency and stronger modeling capabilities, and can independently extract effective feature structures from complex spectral data for learning. The purpose of this paper is to establish a soil nutrient spectrum prediction model with higher efficiency, higher robustness and accuracy, which is of great significance for accelerating the advancement of my country’s agricultural informatization, improving the level of agricultural scientific management and developing my country’s agricultural economy.

Early research found that the soil organic matter can be calculated from the reflectance value of the soil reflectance spectrum, and the response of soil properties can be identified from the spectral characteristics. In 2006, Rossel et al. compared the predictions of various soil concentrations using qualitative analysis values of visible light (VIS) (400 nm–700 nm), near infrared (NIR) (700 nm–2500 nm), and medium infrared (MIR) (2500 nm–5000 nm), demonstrating that soil analysis and soil information can be obtained more effectively using VIS, NIR and MIR [7]. Later, due to the complexity of vis-NIR spectroscopy, a variety of methods were applied to the pretreatment of soil spectra, such as Savitzky-Golay smoothing, standardization, and normalization methods [8,9]. In 2016, Lin et al. used a combined method of S-G smoothing and scattering correction to process soil spectral data to minimize irrelevant and useless information of the spectrum and increase the correlation between the spectrum and the measured value [10]. By choosing the best combination of preprocessing methods to process soil vis-NIR data, not only can the interference factors be eliminated to the greatest extent, but also the complementary relationship between each preprocessing method can be used to improve the prediction accuracy of the network model. In the existing literature, researchers mostly focus on the preprocessing of spectral data, and there are few proposals and improvements of correlation spectral regression models. A high-performance spectral data modeling method can simplify the preprocessing requirements of spectral data and is also the key to ensuring the accuracy of spectral prediction [11]. With the development of regression prediction, more and more linear regression methods are applied to soil nutrient prediction, such as the principal component regression (PCR) of Chang [12] and the partial least square regression (PLSR) method of McCarty [13]. After that, random forest, genetic algorithm, least squares-support vector machines (LS-SVM) and the Cubist method in machine learning are also used to improve model prediction ability [14,15,16,17]. Because deep neural networks are good at automatically extracting useful feature representations from large amounts of data, they have obvious advantages over shallow models and linear methods in modeling, and have become a hot spot in machine learning research in recent years. In 2015, Veres et al. applied deep learning technology to soil spectroscopy for the first time, proving the feasibility of the CNN model for evaluating certain characteristics of LUCAS soil data [18]. In 2017, Ruder proposed that the use of multi-task models can reduce the risk of overfitting while improving the efficiency of model training [19]. In 2019, Padarian et al. used the convolutional neural network (CNN) model and multi-task CNN network to predict various soil properties based on the LUCAS data set, verifying the effectiveness of multi-task learning in predicting soil properties, but the proposed deep learning method is only suitable for large-scale spectral data set, the prediction result is poor on a small sample [20]. After that, Ndikumana et al. used the spectral data as time series data and input it into the long and short-term memory network (LSTM) for soil prediction, and finally achieved good results. However, before training the model, the article performs PCA linear dimensionality reduction processing on the data, which may cause the loss of non-linear correlation between samples, resulting in the model not being able to fit the data characteristics well [21].

Aiming at the problems of low efficiency and low accuracy of current soil prediction models, this paper proposes a new multi-task model based on near-infrared spectroscopy soil data to simultaneously predict multiple attributes of soil. Since the spectral data presents a non-linear trend with the change of the spectral wavelength, this paper takes the spectral wavelength as the time axis, and the spectral data is a non-stationary time series signal. First, the spectrum signal through the three pre-processing methods of SG smoothing, multi-scattering correction, and centralization to construct a stable spectrum sequence. and the original spectral data is windowed and Fourier transform is used to generate a spectrogram, and multiple input channels are used to construct a dual-stream Multi_CNN network that simultaneously inputs a spectrum sequence and a spectrogram, and realizes multiple inputs and multiple outputs of the model by fusing one-dimensional convolution and two-dimensional convolution. In addition, the model has an adaptive input selection function, and independently selects single input and multiple input based on the characteristics of two different scale soil spectral data sets. Due to the small number of samples and short wavelength range of the Small dataset, it only uses a single input to use the one-dimensional convolutional network of the Multi_CNN model for attribute prediction, while LUCAS dataset selects multiple inputs for prediction based on the complete Multi_CNN model. The results show that the evaluation results of single-input network predicting small sample data are better than traditional machine learning algorithms. For large sample data sets, the evaluation results of the Multi_CNN model are better than the existing new models.

The structure of the article in this article is as follows. The second part introduces the two soil sample spectral data and preprocessing methods involved in the article, as well as the multi-input multi-output network Multi_CNN built. The third part compares the two data sets with different scales, and discusses and analyzes the results. The fourth part summarizes the article.

## 2. Materials and Method

### 2.1. The Soil Dataset

Deep learning methods require a large number of samples to train the network, but soil samples based on large national or even global data sets take a long time to sample, so local soil spectral data sets are generally small. The purpose of this article is to make the most accurate predictions for data sets of different scales. The study uses two soil vis-NIR spectroscopy data sets of different scales for prediction modeling.

The Small dataset selected in this paper is to obtain 180 soil samples from 19 sampling sites in Qingdao’s South District, Shibei District, Laoshan District, Huangdao District, and Jiaozhou City. The sampling points are selected to be consistent in color and vegetation coverage. In the region, keep the soil nutrients at each sampling point relatively uniform, and the soil quality is mainly sandy loam or silt loam. After air drying and sifting the soil samples, DH-2000 (Ocean Optics, Dunedin, FL, USA) was used as the light source to conduct soil nutrient spectrum collection by connecting the QE-65000 (Ocean Optics, Dunedin, FL, USA) spectrometer with Y-type optical fiber. The spectral range of the spectrometer is 200 nm–1100 nm, the sampling interval is set to 1 nm, and the integration time is set to 600 ms. In order to eliminate accidental factors, each soil sample in the study was repeated 5 times, and the average value was used for calculation. The obtained reflectance spectrum is greatly interfered in the beginning and ending period, so only the middle reflectance data from 225 nm to 975 nm is retained, and each sample contains 750-dimensional data. The basic information of each sample is shown in Table A1, Table A2, Table A3, Table A4. After that, the physical and chemical values of soil nutrients are obtained through laboratory methods, and the content of the test is the concentration values of soil total nitrogen (TN), total carbon (TC) and alkali hydrolyzed nitrogen (AN). Among them, the physical and chemical values of TN and TC were directly measured by Perkin-Elmer 2400 (PerkinElmer, Waltham, MA, USA) carbon and nitrogen analyzer, and the soil AN attribute value is measured by alkaline hydrolysis diffusion method. First, sodium hydroxide is used to hydrolyze the sample to make the available nitrogen alkaline hydrolyze into ammonia state, then it is absorbed with boric acid, then titrated with standard acid, and finally the content of alkaline hydrolysis nitrogen is calculated.

The large-scale soil data set LUCAS is composed of 19,036 soil sample data collected from 23 European countries, including cultivated land, grassland, woodland and other land types. The soil samples are quite different [8]. After the soil samples are also air-dried and sieved, they are measured by a FOSS XDS (Foss, Denmark) near-infrared spectrometer. The spectral characteristics are recorded with a spectral resolution of 0.5 nm to obtain 4200-dimensional data with a wavelength range of 400 nm–2500 nm. The LUCAS dataset tested a variety of soil properties, such as coarse debris content, organic carbon, nitrogen, potassium, phosphorus, pH, clay, and so forth, so this article selected three soil properties from the LUCAS data set, including organic carbon (OC), nitrogen (N) and Clay, the basic information of the two data sets is shown in Table 1, which shows the diversity and difference of LUCAS sample data. Figure 1 shows the original soil vis-NIR spectra of the two data sets, where the horizontal axis is the wavelength range and the vertical axis is the absorbance of the soil sample.

### 2.2. Data Preprocessing

In the collection of soil spectra, due to environmental factors such as temperature and humidity, instrument state, manual operation, and uneven physical state of the soil, the obtained spectra may contain interference factors such as noise, scattering, and baseline drift. Noise concealing spectral characteristics will reduce the accuracy of soil carbon content prediction. Therefore, the use of preprocessing methods to process the original spectrum can reduce the interference that affects the analysis results and establish a more accurate prediction model.

At present, there are many kinds of spectral preprocessing methods. According to the effect of preprocessing, this paper adopts three types of algorithms: smoothing, scattering correction and scale scaling to eliminate high-frequency noise interference in spectral signals. The Savitzky-Golay (S-G) smoothing algorithm is essentially a weighted average method. The smoothing point data is obtained by least square fitting of the data in the smoothing window with the method of polynomial fitting, so as to reduce the loss of spectral information in smoothing by weighting. Multivariate Scattering Correction (MSC) is one of the commonly used algorithms for spectral data preprocessing. It can effectively eliminate the spectral difference caused by different scattering levels and the shift and shift of baseline caused by the influence of scattering between samples, so as to enhance the correlation between spectra and data and improve the signal-to-noise ratio of original absorbance spectrum [22]. In addition, in order to eliminate the influence of the difference between the dimensions and the value range between the standards, it is usually necessary to centralize the data in the regression problem. The absorbance of the average spectrum is removed by calculating the data value of the absorbance of each spectrum, thereby deducting the value of absolute spectral absorbance. Figure 2a,d are the spectra of the two data sets after S-G smoothing, which can remove the noise peak while retaining the useful spectral information. After the data in Figure 2b,e is processed by MSC, the degree of spectral overlap becomes higher, which reduces the influence of scattering on the original spectrum. Figure 2c,f shows that the data after the centering is scaled based on the origin, eliminating the interference of size difference and different information structure.

In addition, since this paper uses vis-NIR spectroscopy as time series data, it is possible to decompose signal fragments and do Fourier transform to obtain a spectrum map of soil spectral data. In this experimentr, the Hamming window is used for windowing before the Fourier transformation. The frame length on the Small dataset is determined to be 50, the overlap observation value is 20, and there are 180 pictures in total. The LUCAS dataset contains 19036 pictures, the frame length is determined to be 100, and the overlap observation value is 50. Finally, each sample is represented by a 64 × 64 spectrogram. Figure 3 shows the spectrum of soil sample data randomly selected from the two data sets. The horizontal axis represents the wavelength range, and the vertical axis represents the frequency.

### 2.3. The Multiple-Input Multiple-Output Network

Convolutional Neural Networks (CNN) is a nonlinear model, and its unique convolution and pooling structure can extract essential features from complex input information, so it has excellent model characterization capabilities. It is usually composed of multiple convolutional layers, pooling layers and fully connected layers. The convolutional layer is a linear calculation layer that uses a series of convolution kernels to convolve with the input data. It can reduce the number of parameters of the whole model by taking advantage of the localization and positional independence of the features in the input data. The pooling layer is mainly used for feature dimensionality reduction, compressing the number of data and parameters to reduce over-fitting and improve the fault tolerance of the model. The fully connected layer uses neurons to fit the data distribution and improve the model learning ability. Temporal Convolutional Network (TCN) is an innovative network structure that combines the best practices extracted from convolutional neural networks. It transforms into a model suitable for sequence data by combining one-dimensional full convolution and causal convolution [23]. The residual structure of the TCN model connects causal convolution and dilated convolution, and its structure is shown in Figure 4. It includes two dilated convolutional layers, and WeightNorm and Dropout are added after each layer to achieve regularization. The dilated rate of each dilated convolutional layer increases exponentially with the number of levels, ensuring that the convolution kernel covers all inputs in the effective history information, and also ensuring that the use of deep networks can generate extremely long effective history information.

The multi-task method is a natural choice, and its goal is to obtain predictions for multiple tasks at the same time. The multi-task network mainly includes hidden layers shared between all tasks, as well as maintaining several task-specific output layers, where each output layer is associated with a task. This paper builds a multi-task network structure Multi_CNN, through the fusion of one-dimensional convolution and two-dimensional convolution network, to achieve the model’s multiple input and multiple output. The one-dimensional spectral sequence and two-dimensional spectrogram are used as feature data at the same time, which can better fit the spectral features, thereby improving the prediction accuracy of soil properties.

The first input of the model is mainly preprocessed spectral sequence data. By referring to the time convolution network suitable for time series modeling, a one-dimensional convolution network is built to predict multiple soil attributes, and it is named Multi_CNN_1D. The first layer of a one-dimensional convolutional network is a one-dimensional convolutional layer with 64 filters. ReLU is used as the activation function and the convolution kernel weight is normalized to train the network more stably. After that, the BN layer is added, and the maximum pooling layer is used for the down-sampling operation. The third layer is also the convolutional layer with 128 filters. Then the residual module is constructed, which includes two dilated convolutional layers with dilated rate of 2 and 4, and a one-dimensional convolutional layer with exactly the same parameters. Then the fully connected layer connects all the outputs of the previous layer to all the inputs of the next layer and performs information integration. The spectrogram is trained by using a three-layer two-dimensional convolutional layer and adding a pooling layer in the middle, which can reduce the parameter dimension and prevent network overfitting. Then, the feature data type extracted by the dual-stream CNN is converted through the Flatten layer, and finally the prediction results of the three attributes of the soil are output through the three-branch fully connected layer. The network structure is shown in Figure 5, and the specific parameters of the model are shown in Table 2.

## 3. Results and Discussion

The regression fit and accuracy of the network model to the predicted samples are the most important aspects to measure the performance of the model, reflecting the model’s ability to predict unknown samples after training. In order to verify the effectiveness of the multi-task network proposed in this paper, the performance of the model was investigated from the two aspects of regression fitting degree and prediction accuracy. The most unified and objective evaluation criteria were adopted, including determination coefficient ($R^{2}$), modeling root mean square error (RMSEC), prediction root mean square error (RMSEP) and prediction relative analysis error (RPD). $R^{2}$ reflects the closeness between the measured value and the predicted value. RMSEC and RMSEP respectively reflect the degree of deviation between the actual measured value and the predicted value in the prediction of the training set and the test set. RPD reflects the predictive power of the model built, The larger the $R^{2}$ and RPD obtained at the end, the smaller the RMSE, indicating that the performance of the prediction model is better, and vice versa.

### 3.1. Comparison of Pretreatment Methods

Early experiments proved that a suitable preprocessing method can make the model iterate at a faster convergence rate and improve computational efficiency. In this paper, three single methods of the Savitzky-Golay smoothing, multivariate scattering correction and centralization and four combinations of three methods are used to process the spectrum sequence. Due to the complexity of the spectrum data, the processed spectrum is more messy. In order to see the effect comparison more intuitively, three representative samples are selected in the Small dataset for drawing, as shown in Figure 6. Among them, Figure 6a,b are the spectra of S-G smoothing combined with MSC and centralization, Figure 6c is the spectrum of MSC and centralization, and Figure 6d is the spectrum of S-G smoothing, MSC and centralization.

After that, the Small dataset and LUCAS dataset processed by 3 single preprocessing methods and 4 combinations are input into the Multi_CNN_1D network suitable for time series data modeling for training, and the laboratory measured soil attribute value as true value labels in the network, as the training sample according to the batch sent to the input of network framework. Finally, the model test set evaluation effect diagram in Figure 7 is obtained, where the vertical axis represents $R^{2}$. The closer to 1, the higher the accuracy of prediction. It can be seen from Figure 7 that the three soil attributes in the two data sets have the highest $R^{2}$ after S-G smoothing, MSC and centralization combined processing, which means that the gap between the measured value and the predicted value is the smallest, which proves that the S-G smoothing, MSC and centralized combination methods are the best preprocessing methods for processing the spectral sequences of the two data sets.

### 3.2. Experimental Comparison on Small Dataset

Under the condition of small sample of spectral data and selection of the best preprocessing method, the proposed Multi_CNN_1D model is compared with other methods. In this paper, linear regression method is selected as the model comparison method, including Partial Least Square Regression (PLSR), Random Forest Regression (RFR) and Gradient Boosting Decision Tree (GBR). Considering the problem of too few samples in the small data set, the verification set is not split. Instead, the data set is randomly divided into 125 samples in the training set and 55 samples in the test set according to the ratio of 7:3, and then the Multi_CNN_1D model is input for training. Finally, the prediction sequence corresponding to each soil attribute is obtained. By calculating the deviation between the real value of the training set and the corresponding predicted value of the three soil properties TC, TN and AN, and the real value of the test set and the corresponding predicted value, the training set fitting accuracy of the model and the prediction accuracy of the test set are given quantitatively. Finally, the degree of the advantages and disadvantages of each method is evaluated. The results are shown in Table 3. Where $R_{C}^{2}$ and RMSEC are the results of the training set, $R_{P}^{2}$, RMSEP and RPD are the results of the test set.

Figure 8 is a box plot of model evaluations obtained by performing multiple experiments on the test set using different regression models, where the solid line in the middle represents the median of the model evaluation value $R^{2}$. It can be seen from the Table 3 and Figure 8 that the PLSR method performs best in the linear regression method, and the evaluation results of the Multi_CNN_1D model $R^{2}$ and RPD are better than other linear methods. For the same soil attribute TC, the RMSEP error of the Multi_CNN_1D network were 0.24, 0.74 and 0.99 lower than those of traditional PLSR, RFR and GBR networks. Compared with TN, the RMSEP error is reduced by 0.02, 0.11,0.06. Although $R_{P}^{2}$ and RPD of AN were 0.87 and 2.76, which were slightly lower than those of the other two attributes due to the large difference in AN attribute value, the RMSEP error of AN was reduced by 2.64, 5.22 and 1.21 compared with other methods. This indicates that the accuracy of the deep learning method in predicting soil properties based on vis-NIR spectral data is better than the general linear regression method. In addition, when inputting Small dataset samples into the Multi_CNN network for modeling training, the network will have over-fitting problems. This is because the small data set has too few samples and the wavelength range is short, which is more suitable for simple one-dimensional convolutional networks, not multi-input networks. Therefore, for Small dataset, the prediction result of the single-input network is better than that of the dual-stream network.

Figure 9 shows the comparison of the predicted TC, TN, and AN content values of the test set soil samples with the actual values obtained by laboratory method analysis. It can be clearly seen that the scattered points are closely and evenly distributed on both sides of the regression line, and the predicted values of the three soil properties are positively correlated with the actual values, which proves that the multi-task network proposed in this paper is effective in predicting soil properties with a single input.

### 3.3. Experimental Comparison on LUCAS Dataset

The sampling of the LUCAS data set spans the European continent, and the soil samples are very diverse. In this paper, the 19,036 samples of the LUCAS data set are shuffled and divided into training set, validation set and test set according to the ratio of 6:2:2. The number of samples in the training set is 11,420, and the number of samples in the validation set and test set is 3808. Figure 10 shows the scatter plot of the predicted and actual values of the soil samples on the test set using the Multi_CNN model. It can be seen that the scatter values of the three attributes of OC, N, and Clay are evenly distributed on both sides of the regression line. The results show that for large samples of soil spectral data, the proposed Multi_CNN model can effectively extract the characteristic information of soil vis-NIR spectral data. It has high regression fitting and regression accuracy for training samples, and has better learning ability, and can achieve maximum training through existing data, and at the same time accurately approximate the actual measured value of the training sample.

In order to more intuitively reflect the effectiveness of the proposed network model, based on the LUCAS data set, this paper compares the two proposed models with the existing advanced models, including the CNN and multi-task CNN models proposed by Padarian [20] and Ndikumana [21] LSTM model and traditional PLSR model. Table 4 shows the comparison of the evaluation results of each model.

The results show that the $R_{P}^{2}$ of the proposed Multi_CNN and LSTM network [21] in predicting soil N attributes is 0.91, but RMSEP is relatively reduced by 0.06, and the results are better than the LSTM network [21] in predicting OC and Clay. Compared with CNN_multi [20], the prediction results $R_{P}^{2}$ and RMSEP of the three attributes of OC, N and Clay proposed in this paper are improved a lot. This is because this paper uses spectral data as time series data to learn the short-term and long-term dependence of sample data, and fuse one-dimensional convolution and two-dimensional convolution makes the feature fit better and the prediction accuracy more accurate. In addition, the prediction effect of the three attributes of the Multi_CNN network is also higher than that of the proposed single-input Multi_CNN_1D network. Therefore, the self-adaptive Multi_CNN network built in this paper can obtain better prediction results for different scale data sets.

In order to further verify the effectiveness of the proposed algorithm, this paper selects Qingdao soil spectral data measured in different periods, which contains 500 data samples, and the selected soil attributes are also TC, TN, and AN. The small samples are preprocessed and input into the adaptive network Multi_CNN. Finally, the modulus evaluation parameters $R_{P}^{2}$ of the three attributes are 0.91, 0.98 and 0.95, and the RMSEP is 0.17, 0.02, and 2.21. The three soil attributes are shown in the Figure 11. A scatter plot of the predicted value and the actual value of the attribute obtained by laboratory method analysis. It can be seen that the scattered points are evenly distributed on both sides of the regression line, which proves that the Multi_CNN network has high predictive ability and generalization ability.

## 4. Conclusions

This paper proposes a new intelligent network architecture for simultaneous soil multi-attribute prediction in the same task network. The proposed framework is based on soil vis-NIR spectral signals, and a dual-stream convolutional network is built to predict various characteristics of soil. The spectral signal was processed by the combination of pretreatment and the conversion of the original data to the spectral map, which made the soil characteristic information extracted by the network more detailed. In addition, this paper discusses the predictive ability of soil data set networks based on different scales. Due to the small number of samples and short wavelength range of the Small dataset, it is more suitable for one-dimensional convolution input, but not for the complex network structure with multiple inputs and outputs. However, for the large-scale LUCAS dataset, the multi-input and multi-output network significantly improves the prediction accuracy, and the results are better than the existing methods. This paper fully proves the feasibility and accuracy of multi-task network in soil attribute prediction.

## Figures and Tables

**Figure 1 sensors-20-06271-f001:**
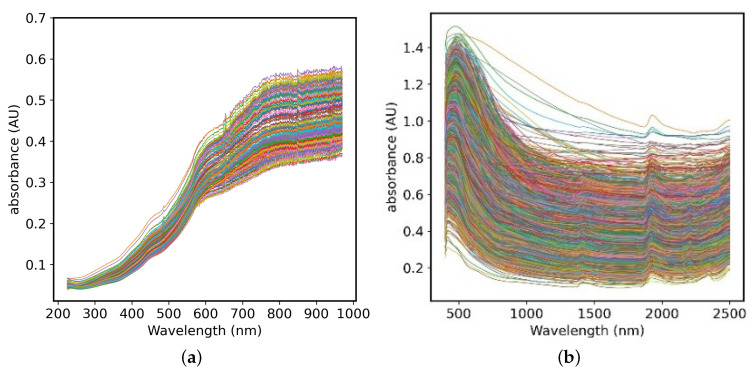
Original spectrogram of (**a**) Small dataset (**b**) LUCAS dataset.

**Figure 2 sensors-20-06271-f002:**
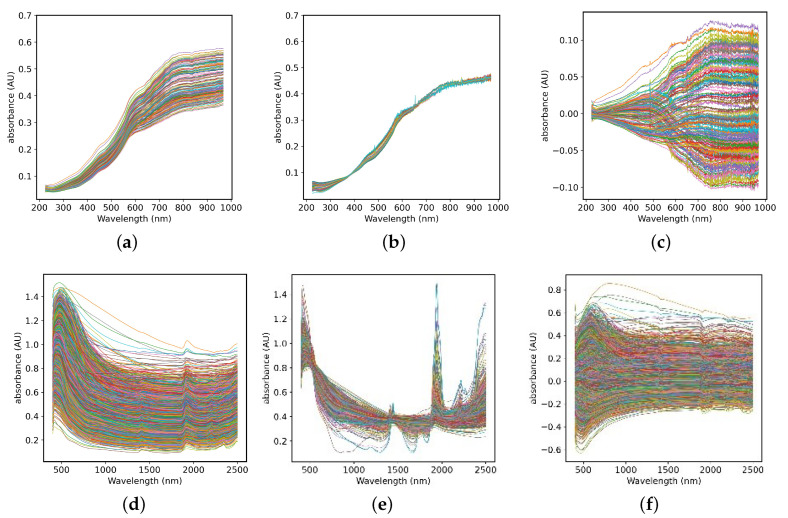
The Spectra of the small dataset (**a**–**c**) and LUCAS dataset (**d**–**f**) preprocessed by Savitzky-Golay (S-G), Multivariate Scattering Correction (MSC) and Centralization methods.

**Figure 3 sensors-20-06271-f003:**
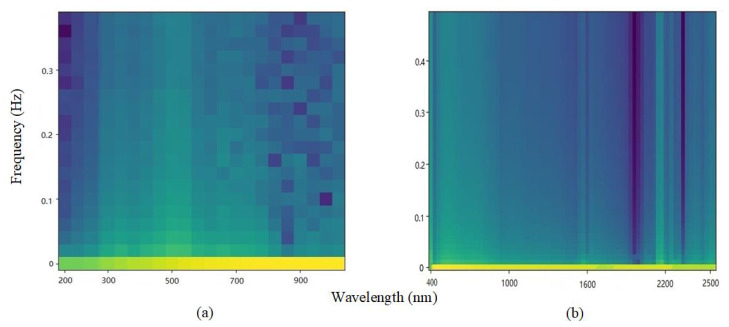
Spectrograms of samples of (**a**) Small dataset (**b**) LUCAS dataset. (The yellower the color represents the greater the spectral frequency density at this wavelength, the bluer the color, the lower the density.)

**Figure 4 sensors-20-06271-f004:**
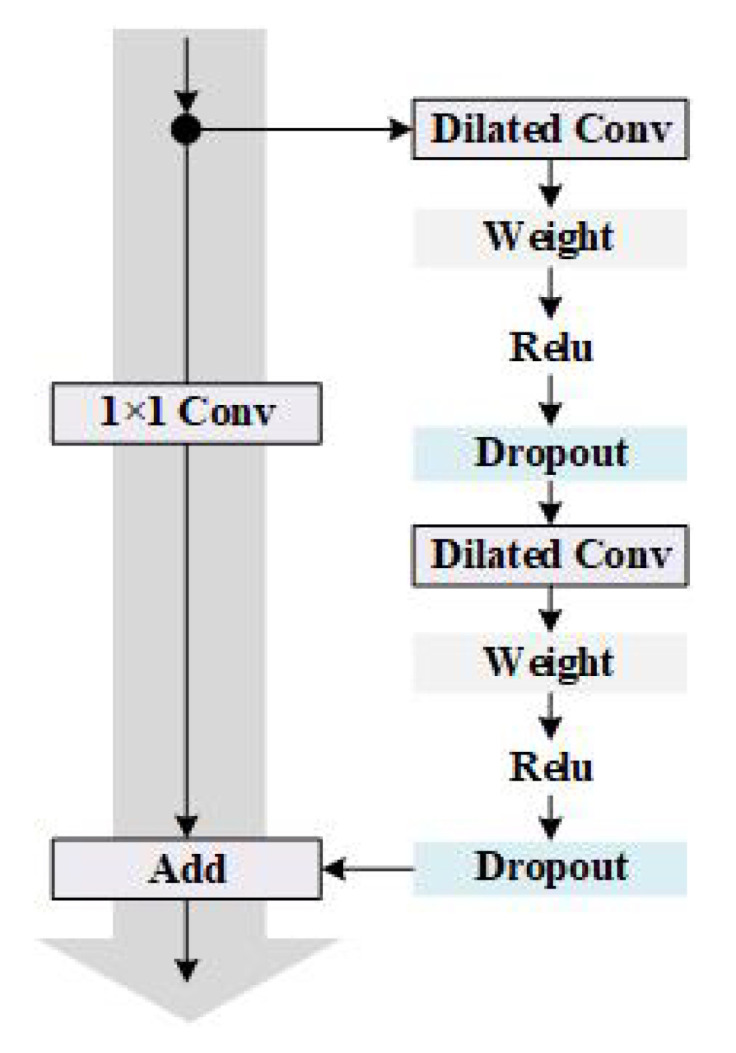
The residual module of Temporal Convolutional Network (TCN).

**Figure 5 sensors-20-06271-f005:**
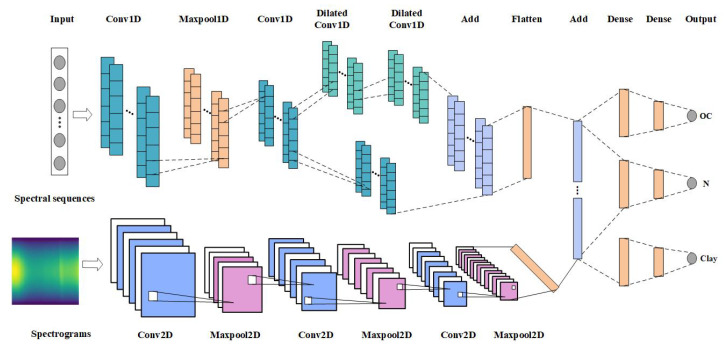
The Multi_CNN network structure.

**Figure 6 sensors-20-06271-f006:**
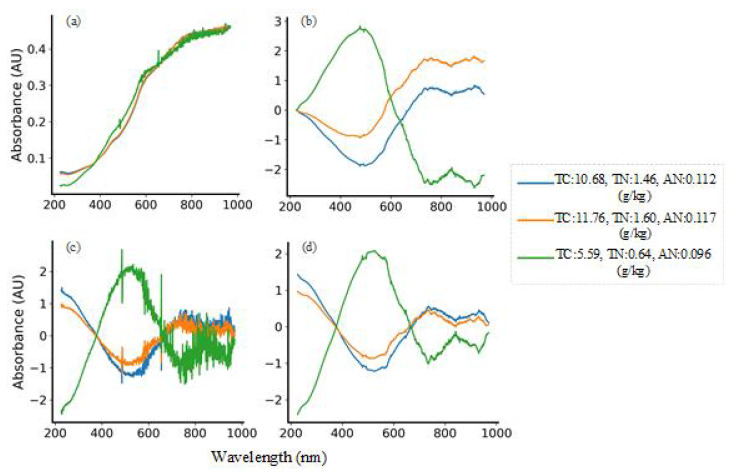
Preprocessed spectra of (**a**) S-G+MSC (**b**) S-G+Centralization (**c**) MSC+Centralization (**d**) S-G+MSC+Centralization.

**Figure 7 sensors-20-06271-f007:**
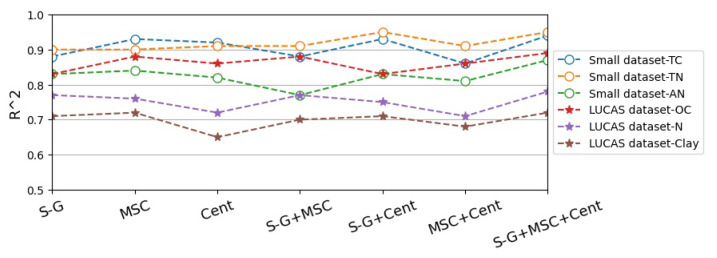
Comparison diagram of preprocessing effect on test set. (S-G refers to Savitzky-Golay smoothing algorithm, MSC refers to Multivariate Correction, Cent refers to Centralization methods.)

**Figure 8 sensors-20-06271-f008:**
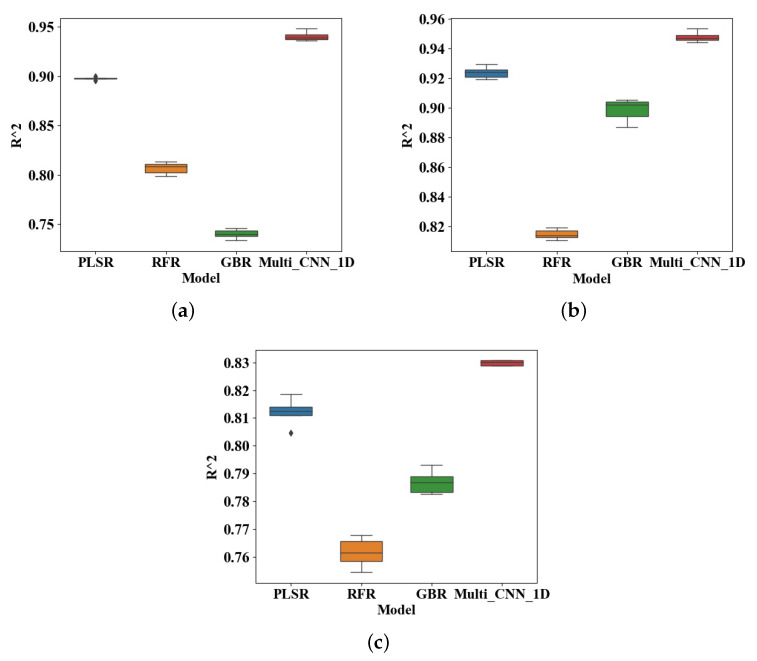
$R^{2}$ comparison box diagram of predicted values of (**a**) Total Carbon (**b**) Total Nitrogen (**c**) Alkaline Nitrogen.

**Figure 9 sensors-20-06271-f009:**
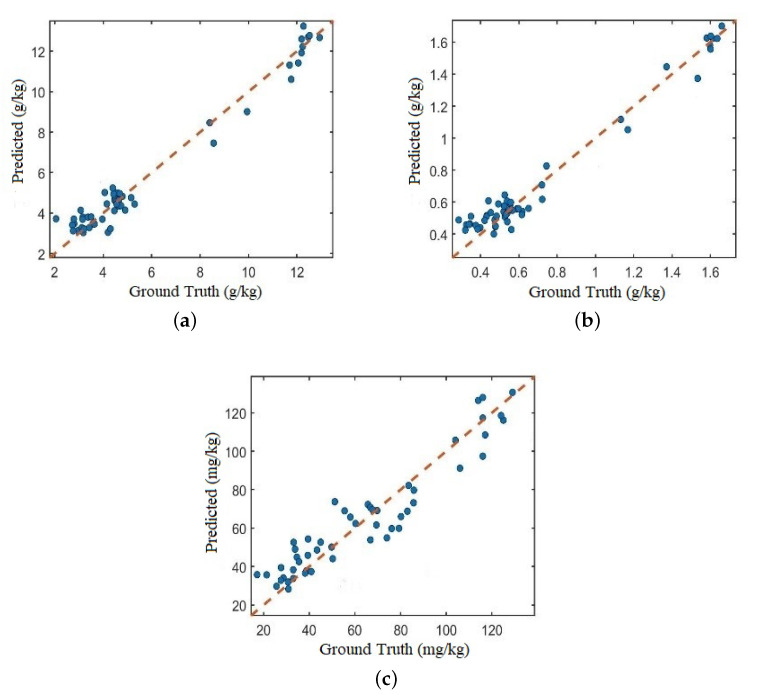
Actual vs Predicted values of Proposed framework (**a**) Total Carbon (**b**) Total Nitrogen (**c**) Alkaline Nitrogen.

**Figure 10 sensors-20-06271-f010:**
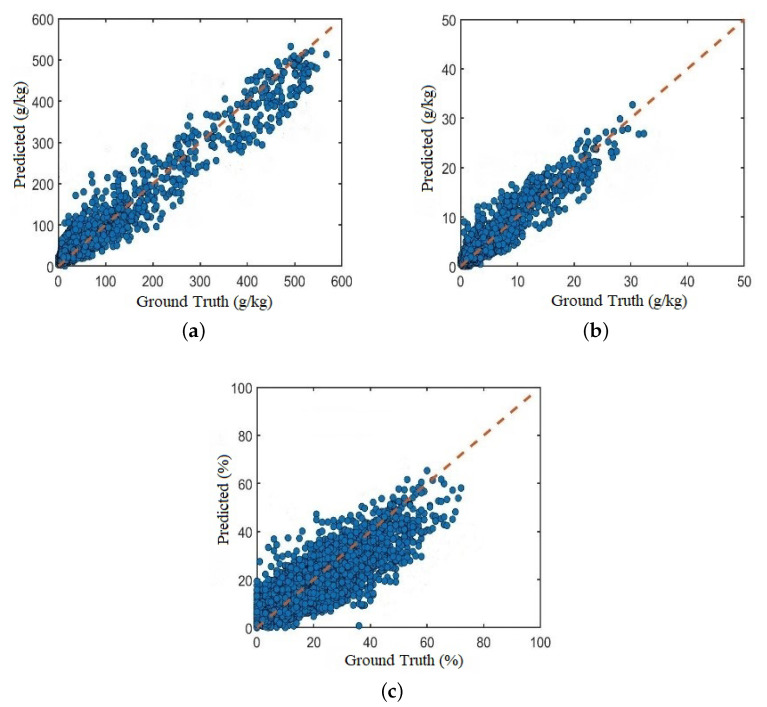
Actual vs Predicted values of Multi_CNN model (**a**) Organic Carbon (**b**) Nitrogen (**c**) Clay.

**Figure 11 sensors-20-06271-f011:**
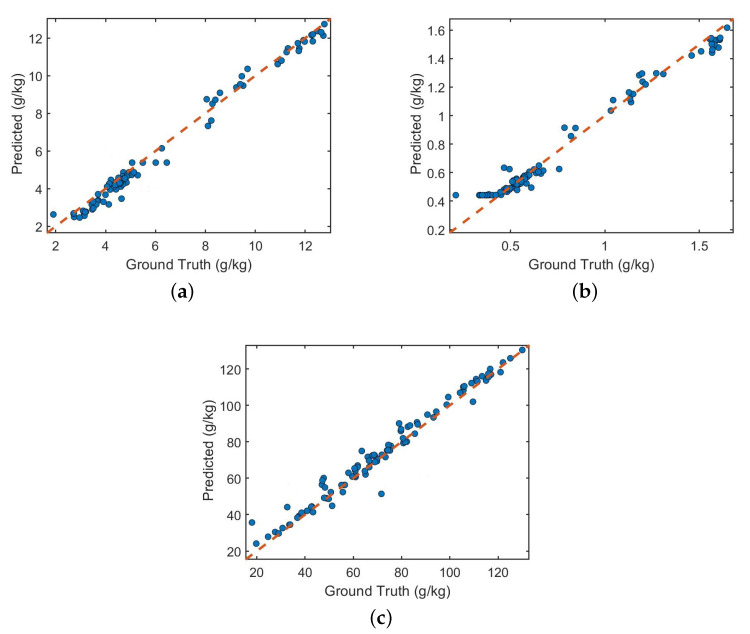
Actual vs. Predicted values of Proposed framework (**a**) Total Carbon (**b**) Total Nitrogen (**c**) Alkaline Nitrogen.

**Table 1 sensors-20-06271-t001:** Basic information of soil data set.

		Minimum	Maximum	Mean	Median	St.Dev.
Small dataset	TC (g·kg−1)	1.90	13.40	6.21	4.62	3.42
	TN (g·kg−1)	0.21	1.74	0.80	0.56	0.46
	AN (mg·kg−1)	17.2	160.00	69.51	62.15	33.18
LUCAS dataset	OC (g·kg−1)	0.00	586.80	50.00	20.80	91.31
	N (g·kg−1)	0.00	38.60	2.92	1.70	3.76
	Clay (g·kg−1)	0.00	79.00	18.88	17.00	13.00

**Table 2 sensors-20-06271-t002:** Multi_CNN network specific parameter settings.

Layer	Kernel Size	Filters	Layer	Kernel Size	Filters
Conv1D	3	64	Conv2D	5 × 5	64
Maxpool1D	5	-	Maxpool2D	2 × 2	-
Conv1D	3	128	Conv2D	3 × 3	128
Conv1D	3,	64	Maxpool2D	2 × 2	-
	dr = 2				
Conv1D	3,	64	Conv2D	3 × 3	256
	dr = 4				
Conv1D	3	64	Maxpool2D	2 × 2	-
FC1	-	128			
FC2	-	64			
FC3	-	1			

**Table 3 sensors-20-06271-t003:** Comparison of evaluation indexes of each model based on Small dataset.

Attributes		PLSR	RFR	GBR	Multi_CNN_1D
TC	$R_{C}^{2}$	0.96	0.97	0.97	0.99
	RMSEC	0.74	0.50	0.57	0.23
	$R_{P}^{2}$	0.89	0.81	0.73	0.94
	RMSEP	1.04	1.54	1.79	0.80
	RPD	3.01	2.30	1.95	4.23
TN	$R_{C}^{2}$	0.98	0.96	0.97	0.99
	RMSEC	0.07	0.10	0.09	0.04
	$R_{P}^{2}$	0.93	0.82	0.90	0.95
	RMSEP	0.11	0.20	0.15	0.09
	RPD	3.80	2.34	3.20	4.71
AN	$R_{C}^{2}$	0.99	0.96	0.89	0.95
	RMSEC	3.59	5.98	11.86	7.36
	$R_{P}^{2}$	0.81	0.76	0.83	0.87
	RMSEP	14.26	16.84	12.83	11.62
	RPD	2.31	2.04	2.43	2.76

**Table 4 sensors-20-06271-t004:** Comparison of model evaluations on the LUCAS dataset.

Attributes		Multi_CNN_1D	Multi_CNN	CNN [20]	CNN_multi [20]	LSTM [21]	PLSR
OC	$R_{P}^{2}$	0.89	0.95	0.88	0.69	0.94	0.54
	RMSEP	29.49	22.69	32.14	31.86	23.25	68.12
N	$R_{P}^{2}$	0.78	0.91	0.83	0.60	0.91	0.55
	RMSEP	1.76	1.09	1.54	1.59	1.15	1.73
Clay	$R_{P}^{2}$	0.72	0.83	0.70	0.68	0.80	0.50
	RMSEP	7.17	5.53	7.55	7.78	5.95	8.93

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
