# Peer review of "Simultaneous Prediction of Soil Properties Using Multi_CNN Model"

_sensors, 2020, doi:10.3390/s20216271_

Round 1

Reviewer 1 Report

Comments:

This paper proposes a soil multi-attribute intelligent prediction method based on convolutional neural network. The results are calibrated and validated by two different datasets. This work is meaning for improving the predicted accuracy of soil properties from soil spectral data.

Some suggestion is given as follows:

  • In Figure 1, why small dataset and LUCAS dataset has different value range and curve shape? What is their units? Their value should be smaller than one if their value represents the reflectance of soil samples. And it seems right for the curve shape of small dataset.
  • How much samples for training?And how much for validation? Please clarify.
  • Line 30: please check this line for “.Loss”
  • Figure 3: please add the legend of Figure 3, and we can get the meanings of the color.
  • The 2nd Line 130: this sentence should be checked. “Fig 2. Spectra of the two data sets preprocessed by S-G, MSC and Centralization. “
  • Line 178-179: “The model training process will be pre-processed The later soil vis-NIR spectrum data is used as the input signal….”, punctuation missing?
  • Figure 7: the four subplots of Figure 7 represent what? And (c) and (d) seems like before and after smoothing? What about (a) and (b)?
  • Figure 8: what is the meanings for color bar? Please add the legend.
  • Figure 9 seems no contribution for the result of this study
  • Two datasets (small dataset and LUCAS dataset) has different range of wavelength. This can lead to some difference in the accuracy of estimated soil properties. Please explain the reason for select two different dataset.

Author Response

Dear reviewer,

Many thanks for these valuable suggestions. According to your suggestions, we revised the manuscript. The changes we made are as follows:

Point 1: In Figure 1, why small dataset and LUCAS dataset has different value range and curve shape? What is their units? Their value should be smaller than one if their value represents the reflectance of soil samples. And it seems right for the curve shape of small dataset.

Response 1: Because the Small dataset and the LUCAS dataset are collected using different instruments, the number of samples and the wavelength range are also different, so the curve shapes of the two data sets are different. The horizontal axis in Figure 1 represents the wavelength range and the unit is nm. The ordinate is the absorbance of the spectrum, generally there is no unit.

Point 2: How much samples for training?And how much for validation? Please clarify.

Response 2: Considering the problem of too few samples in the small data set, the verification set is not split, but the 180 samples in the small data set are randomly divided into 125 samples in the training set and 55 samples in the test set according to the ratio of 7:3 after being scrambled. The paper has been modified on page 9, line 250-252.

The 19036 samples in LUCAS divide the data set according to the ratio of 6:2:2, among which the training set samples are 11420, and the verification set and test set samples are 3808. The changes in the paper are on page 11, lines 284-286.

Point 3: Line 30: please check this line for “.Loss”

Response 3: The original .Loss in line 30 is a grammatical error caused by my carelessness, which has been deleted from the paper.

Point 4: Figure 3: please add the legend of Figure 3, and we can get the meanings of the color.

Response 4: Figure 4 has been modified in the paper, where the horizontal axis represents the wavelength range and the vertical axis represents the frequency. The yellower the color represents the greater the spectral frequency density at this wavelength, the bluer the color, the lower the density.

Point 5: The 2nd Line 130: this sentence should be checked. “Fig 2. Spectra of the two data sets preprocessed by S-G, MSC and Centralization. “

Response 5: The 130th line in the previous article is the legend of the Figure 2, because I accidentally put it at the beginning of the paper. The original sentence on line 130 in the paper has been deleted.

Point 6: Line 178-179: “The model training process will be pre-processed The later soil vis-NIR spectrum data is used as the input signal….”, punctuation missing?

Response 6: The original line 178-179 punctuation was accidentally written incorrectly. Due to the unreasonable structure of the previous article, the current section 2.3 has been rewritten.

Point 7: Figure 7: the four subplots of Figure 7 represent what? And (c) and (d) seems like before and after smoothing? What about (a) and (b)?

Response 7: The previous Figure 7 has now been changed to Figure 6, and the legend has been added. Among them, (a), (b) are the spectra of S-G smoothing combined with MSC and centralization, (c) is the spectrum of MSC and centralization, and (d) is the spectrum of S-G smoothing, MSC and centralization.

Point 8: Figure 8: what is the meanings for color bar? Please add the legend.

Response 8: The previous Figure 8 has been changed to Figure 7. In order to show the pre-processing effect of the three attributes of the two data sets more intuitively, Figure 7 has been revised. Figure 7 shows the R2 results of the test set after the data of the Small dataset and the LUCAS dataset are input into the Multi_CNN_1D network through these pretreatment methods.

Point 9: Figure 9 seems no contribution for the result of this study

Response 9: The previous Figure 9 is to show the spectral data of the two data sets after pretreatment, which does not make much contribution to the paper. The picture has been deleted according to your opinion.

Point 10: Two datasets (small dataset and LUCAS dataset) has different range of wavelength. This can lead to some difference in the accuracy of estimated soil properties. Please explain the reason for select two different dataset.

Response 10: Based on large national or even global data sets, the sampling range of soil samples is wide and time-consuming and consumes a lot of resources. Therefore, the local soil spectral data sets collected by themselves are generally small. In order to make the most accurate predictions for data sets of different scales, this paper proposes an adaptive soil multi-attribute prediction network based on two soil vis-NIR spectroscopy data sets of different scales. The Small dataset independently select a single channel input Multi_CNN_1D, and select multiple input Multi_CNN for large LUCAS dataset, and the effectiveness of the algorithm is verified through experiments. And it has been explained in lines 100-104 on page 3 of the paper.

In response to your suggestions, I have highlighted the modified content and pictures in the attachment "sensors-964252-r1", please check.

Kind Regards,

Ruixue Li

Reviewer 2 Report

The paper title “Simultaneous prediction of soil properties using Multi_CNN model” present interesting results concerning the soil multi-attribute intelligent prediction method based on convolutional neural network.  In my opinion the paper could be accepted with some improvements.

In the abstract the codes TN, TC, AN, OC and N must be identified.

In “Keywords: Soil vis-NIR spectroscopy “ I think is “Keywords: Soil, vis-NIR spectroscopy”

The section “2. Related work” must be included in the section “1. Introduction”

Section 3 and 4 must be only one, named material and methods with the appropriates subsection.

Section 5. Experimental results and discussion must be 5. results and discussion.

About the soil samples more explanation was needed, because the author used 180 samples but no explanation about the diversity of soils or their classification is missing. More information about samples locations and characteristics is needed

Between the number and the units must have a space, please revise in all document (Ex. Line 103)

Revise the sentence in line 106

Revise the units in line 112

The spectral data acquisition is not clear, more information about the way of samples was measured as well the condition, was needed. For the equipment, the information about model, supplier and country are needed.

The soil analyses methodologies and equipment identification are need.

All Tables and Figures mut be self-readable, so included in the legend the meaning of the abbreviation.

Is not necessary present the equation of the calculation of R2, RMSE and RPD

In line 234 is SG or S-G?

The results must be better discussed concerning the observed values of RPD and RMSE, because, in my opinion some of them are not very good.

In Table 3 is better use a line to separate TC, TN and AN results, the way it are is a little confuse.

Figure 12 b and c the xx and yy axes must be in accordance with the plotted values.

For me the models seem not be very good and the variability is higher in some cases and must be better explained.

Hy the author does not make an external validation?

Why the variable measured in soil for LUCAS data set is different of the used in the other group?

Conclusion: “… the convolutional neural network is used for soil concentration prediction ….” Soil Concentration?? what does this mean?

Conclusion must be rewrite, they are very confuse and difficult to read.

Author Response

Dear reviewer,

Many thanks for these valuable suggestions. According to your suggestions, we revised the manuscript. The changes we made are as follows:

Point 1: In the abstract the codes TN, TC, AN, OC and N must be identified.

Response 1: TN, TC, AN, OC and N have been defined in the abstract, and the modification location has been marked in line 11-13 of the attachment.

Point 2: In “Keywords: Soil vis-NIR spectroscopy “ I think is “Keywords: Soil, vis-NIR spectroscopy”

Response 2: Following your suggestions, the keywords have been modified to Soil, vis-NIR spectroscopy, and the modification location has been marked in the keywords.

Point 3: The section “2. Related work” must be included in the section “1. Introduction”

Response 3: Following your suggestions, "2. Related work" was put in "1. Introduction", which has been modified in the attachment.

Point 4: Section 3 and 4 must be only one, named material and methods with the appropriates subsection.

Response 4: Combine the previous section 3 and Section 4 into the current section 2 "Materials and Methods", and introduce the data set, pre-processing method and network model used in this article in sections.

Point 5: Section 5. Experimental results and discussion must be 5. results and discussion.

Response 5: The previous section ‘5 Experimental results and discussion’ has been changed to ‘3. Results and discussion’

Point 6: About the soil samples more explanation was needed, because the author used 180 samples but no explanation about the diversity of soils or their classification is missing. More information about samples locations and characteristics is needed

Response 6: The Small dataset used in this paper includes the sampling locations of 180 soil samples for the Qingdao's South District, Shibei District, Shan District, Huangdao District, and Jiaozhou City. It has been explained in line 105-107 on page 3 of the paper. The soil properties of each sample, TC, TN, AN and the Minimum, Maximum, Mean and Median of spectral signals have been explained in Table A1-A4 in the appendix.

Point 7: Between the number and the units must have a space, please revise in all document (Ex. Line 103)

Response 7: All the Spaces between the numbers and units in the paper have been modified.

Point 8: Revise the sentence in line 106

Response 8: The sentence in line 106 has been modified and marked in line 118 on page 3 of the attachment.

Point 9: Revise the units in line 112

Response 9: The unit on line 112 of the original article has been modified on line 129 on page 3 of the attachment.

Point 10: The spectral data acquisition is not clear, more information about the way of samples was measured as well the condition, was needed. For the equipment, the information about model, supplier and country are needed.

Response 10: The spectral data collection information of the Small dataset is in line 107-116 on page 3 of the attachment, and the data collection information of the LUCAS dataset is in line 127-129 of the attachment.

Point 11: The soil analyses methodologies and equipment identification are need.

Response 11: The laboratory soil TN and TC attribute values were directly determined by by Perkin-Elmer 2400 carbon and nitrogen analyzer, and soil AN attribute values were determined by alkali hydrolysis and diffusion method, which were explained in line 119-124 on page 3 of the attachment.

Point 12: All Tables and Figures mut be self-readable, so included in the legend the meaning of the abbreviation.

Response 12: All abbreviations in the images and tables in this attachment have been indicated in the preceding paragraph or in the title of the image.

Point 13: Is not necessary present the equation of the calculation of R2, RMSE and RPD

Response 13: According to your suggestion, the calculation formulas of R2, RMSE and RPD in the attachment have been deleted.

Point 14: In line 234 is SG or S-G?

Response 14: The Savitzky-Golay smoothing algorithm in the paper has been shortened to S-G smoothing.

Point 15: The results must be better discussed concerning the observed values of RPD and RMSE, because, in my opinion some of them are not very good.

Response 15: The discussion of RMSE and RPD in small data is shown in lines 262-269 on page 9 and 11. The discussion of LUCAS 'prediction results is in 299-302 on page 11 of the attachment.

Point 16: In Table 3 is better use a line to separate TC, TN and AN results, the way it are is a little confuse.

Response 16: According to your suggestion, the TC, TN and AN results in Table 3 have been separated by horizontal lines.

Point 17: Figure 12 b and c the xx and yy axes must be in accordance with the plotted values.

Response 17: The original Figure 12 has been modified to the present Figure 10 in the attachment.

Point 18: For me the models seem not be very good and the variability is higher in some cases and must be better explained.

Response 18: This paper builds an adaptive network model. The Small data sets and large LUCS dataset can automatically select single input Multi_CNN_1D and multiple input Multi_CNN  for soil attribute prediction according to different characteristics of the data set. After experimental verification, the prediction results of the two data sets in this paper are higher than the linear method and the existing new network, which proves the validity and accuracy of the model prediction.

Point 19: Why the author does not make an external validation?

Response 19: For external verification, my understanding is to use an external data set to verify the model proposed in this paper. For large data sets, the public data set used in this article is the LUCAS2009 dataset. The LUCAS2015 dataset released by its organization only has soil attribute values, and the vis-NIR spectroscopy data has not been released yet. In addition, this article found an ICRAF-ISRIC soil spectral data set from 58 countries/regions in Africa, Asia, North America and South America, but the downloaded soil attribute tags are confused and the spectral data cannot correspond. Therefore, this paper uses a laboratory test data set containing 500 data samples to verify the model of this paper. Through experiments, Figure 11 fully verifies that the Multi_CNN network proposed in this paper has high accuracy and model generalization ability. And it has been stated in the attachment in lines 309-316 on page 12.

Point 20: Why the variable measured in soil for LUCAS data set is different of the used in the other group?

Response 20: The soil properties in the Small dataset used in this article are total carbon, total nitrogen, and alkaline hydrolyzed nitrogen. The LUCAS dataset tested a variety of soil properties, such as organic carbon, coarse debris content, nitrogen, potassium, phosphorus, pH, clay, etc., but the TC and AN in the Small dataset are not included, so this article cannot select the same variables in the LUCAS dataset as the Small dataset. it has been explained in line 130-132 on page 3 of the attachment.

Point 21: Conclusion: “… the convolutional neural network is used for soil concentration prediction ….” Soil Concentration?? what does this mean?

Response 21: ‘Soil Concentration’ should be the ‘Soil attribute’ and it has been modified in the conclusion of the attachment.

Point 22: Conclusion must be rewrite, they are very confuse and difficult to read.

Response 22: The conclusion has been rewritten and marked in the attachment.

In response to your suggestions, I have highlighted the modified content and pictures in the attachment "sensors-964252-r2", please check.

Kind Regards,

Ruixue Li

Round 2

Reviewer 1 Report

  1. Small dataset and LUCAS dataset were used here with different wavelength range. The wavelength range for Small dataset is ~225~975nm, and its wavelength range of LUCAS dataset is ~400-2500nm. The reason why figure 1 (a) shows the data of about 400-800nm.
  2. The value range of absorbance differs from Small dataset and LUCAS dataset for the same wavelength range (400~800), What causes this result?

Author Response

Dear reviewer,

Many thanks for these valuable suggestions. According to your suggestions, we revised the manuscript. The changes we made are as follows:

Point 1: Small dataset and LUCAS dataset were used here with different wavelength range. The wavelength range for Small dataset is ~225~975nm, and its wavelength range of LUCAS dataset is ~400-2500nm. The reason why figure 1 (a) shows the data of about 400-800nm.

Response 1: The wavelength range of the Small dataset is 225nm ~ 975nm. Figure 1 (a) is my accidental error in setting the axis range, which has been modified in Figure 1 (a) on the page 4 of the attachment.

Point 1: The value range of absorbance differs from Small dataset and LUCAS dataset for the same wavelength range (400~800), What causes this result?

Response 2: Due to the different soil textures of the two data sets, the different equipment used, and the influence of interferences, the absorbance of the two data sets at the same wavelength is different.

In response to your suggestions, I have highlighted the modified content in the attachment 'sensors-964252-r1', please check.

Kind Regards,

Ruixue Li
